# Nestin Modulates Airway Smooth Muscle Cell Migration by Affecting Spatial Rearrangement of Vimentin Network and Focal Adhesion Assembly

**DOI:** 10.3390/cells11193047

**Published:** 2022-09-29

**Authors:** Ruping Wang, Sakeeb Khan, Guoning Liao, Yidi Wu, Dale D. Tang

**Affiliations:** Department of Molecular and Cellular Physiology, Albany Medical College, Albany, NY 12208, USA

**Keywords:** nestin, intermediate filament protein, smooth muscle, migration

## Abstract

Airway smooth muscle cell migration plays a role in the progression of airway remodeling, a hallmark of allergic asthma. However, the mechanisms that regulate cell migration are not yet entirely understood. Nestin is a class VI intermediate filament protein that is involved in the proliferation/regeneration of neurons, cancer cells, and skeletal muscle. Its role in cell migration is not fully understood. Here, nestin knockdown (KD) inhibited the migration of human airway smooth muscle cells. Using confocal microscopy and the Imaris software, we found that nestin KD attenuated focal adhesion sizes during cell spreading. Moreover, polo-like kinase 1 (Plk1) and vimentin phosphorylation at Ser-56 have been previously shown to affect focal adhesion assembly. Here, nestin KD reduced Plk1 phosphorylation at Thr-210 (an indication of Plk1 activation), vimentin phosphorylation at Ser-56, the contacts of vimentin filaments to paxillin, and the morphology of focal adhesions. Moreover, the expression of vimentin phosphorylation-mimic mutant S56D (aspartic acid substitution at Ser-56) rescued the migration, vimentin reorganization, and focal adhesion size of nestin KD cells. Together, our results suggest that nestin promotes smooth muscle cell migration. Mechanistically, nestin regulates Plk1 phosphorylation, which mediates vimenitn phosphorylation, the connection of vimentin filaments with paxillin, and focal adhesion assembly.

## 1. Introduction

Airway smooth muscle (ASM) cell migration plays an essential role in regulating development and repair of the respiratory system. Moreover, ASM cell migration contributes to the progression of airway remodeling, a hallmark of allergic asthma. Asthmatic human ASM cells migrate faster than nonasthmatic cells [1,2]. The luminal border of airway smooth muscle of patients with severe asthma is closer to the epithelium compared to patients with mild asthma [3]. However, the mechanisms that regulate cell migration are not yet fully understood. 

In response to guidance signals in their environment, cells extend the membrane to form protrusion called lamellipodia at the front. New focal adhesions are assembled in cell protrusion to strengthen their attachment to the extracellular matrix (ECM). Subsequently, myosin activity increases to promote stress fiber formation and the retraction of the rear. Finally, focal adhesions at the rear are disassembled to allow whole cell body to move forward [2,4]. 

Nestin is a class VI intermediate filament (IF) protein that was first described as a neuronal stem/progenitor cell marker during central nerve system development [5]. Subsequently, nestin was found in cancer cells, including lung adenocarcinoma [6] and regenerative skeletal muscle [7]. Nestin contains a short N-terminus and a long C-terminus, which makes it impossible to form nestin homopolymers. Nestin interacts with other IF proteins including vimentin to assemble heterodimers and mixed polymers [7]. Nestin is involved in the proliferation of neurons and cancer cells and the regeneration of skeletal muscle [6,7,8]. The role of nestin in cancer cell migration is complex, as its effects vary depending on cell types. The presence of nestin promotes the migration of pancreatic cancer cells [9]. In contrast, nestin has no role in regulating the migration of prostate cancer PC-3 cell line [10]. Moreover, the cellular mechanisms by which nestin regulates migration are largely unknown. 

Focal adhesions are integrin-containing and multi-protein structures that provide physical links between the actin cytoskeleton and the ECM. In response to changes in their environment, integrins engage with the ECM, which promotes integrin clustering and focal adhesion assembly. Focal adhesion formation in lamellipodia strengthens the cell-ECM linkage, which eventually allows cells to crawl.

Vimentin is a type III IF protein which is expressed in various cell types, including smooth muscle cells [2,11,12]. The vimentin network undergoes reorganization during cell migration and plays a role in regulating migration [11,13]. The rearrangement of the vimentin network is regulated by its phosphorylation at Ser-56 [14,15]. 

Polo-like kinase 1 (Plk1) is a serine/threonine protein kinase that plays a role in mitosis [16,17,18], proliferation [19,20], and contraction [14,21]. In addition, Plk1 is also involved in the regulation of cell migration and focal adhesion assembly [15,22]. 

Here, we reveal a previously unknown mechanism by which nestin regulates cell migration. Nestin regulates Plk1 activation, which affects the spatial organization of vimentin filaments, focal adhesion assembly, and migration. 

## 2. Materials and Methods

### 2.1. Cell Culture 

Human ASM cells were prepared from human bronchi and adjacent tracheas obtained from the International Institute for the Advancement of Medicine as previously described [23,24,25,26,27,28], with studies herein approved by the Albany Medical College Committee on Research Involving Human Subjects. The donor human lungs used to procure tissue and cells were not suitable for transplant, and not identifiable, thus studies were determined to be *Not Human Subjects Research*. Smooth muscle cells passage 3–10 were used for the studies, as per [15,29,30]. The purity of HASM was determined by immunostaining for smooth muscle α-actin. Nearly 100% of these cells expressed α-actin [15]. Primary cells from three non-asthmatic donors were used for most experiments. In some cases, duplicate or triplicate experiments from cells of one donor were used for analysis.

### 2.2. Immunoblot Analysis and Coimmunoprecipitation

Western blotting of cell lysis and coimmunoprecipitation were performed using the experimental procedures as previously described [14,15,18,24,31,32]. Antibodies used were anti-nestin (1:500, Fisher/Invitrogen, Waltham, MA, USA. #PIPA511887, L/N SH2420723H and Santa Cruz Biotechnology, Dallas, TX, USA, sc-23927, I0418, L1818, J1519), anti-GAPDH (1:1500, Ambion #AM4300, L/N 1311029), anti-phospho-myosin light chain (Ser-19, Santa Cruz Biotechnology, 1:500), anti-myosin light chain (1:1000, a gift of Dr. Gunst), anti-vimentin (BD Biosciences, San Jose, CA, USA, #550515, L/N 3214517, 1:10,000), anti-Plk1 (Cell Signaling, Danvers, MA, USA, #4535S, L/N 2), anti-phospho-Plk1 (Thr-210, Cell Signaling, #9062 S), anti-c-Abl (Cell Signaling, #2862S, L/N 15), and anti-cortactin (Santa Cruz Biotechnology, #sc-55579, L/N E0417). Phospho-vimentin (Ser56) antibody (1:500) was produced as previously described [14,33]. Anti-nestin, anti-c-Abl, and anti-Plk1 were validated by using corresponding KD cells. Other antibodies were validated by examining molecular weight of detected bands. Finally, vendors have provided datasheet to show that antibodies were validated by positive controls. The levels of proteins were quantified by scanning densitometry of immunoblots (Fuji Multi Gauge Software, Japan or GE IQTL software, Chicago, IL, USA). The luminescent signals from all immunoblots were within the linear range. 

### 2.3. Immunofluorescence Microscopy

Cells were plated in dishes containing collagen-coated coverslips and cultured in a CO_2_ incubator for desired times, followed by fixation and permeabilization as previously described [23,33,34]. These cells were immunofluorescently stained using primary antibody followed by appropriate secondary antibody conjugated to Alexa-488 or Alexa-555 (Invitrogen, Waltham, MA, USA). Primary antibodies used were: anti-nestin (1:25, Invitrogen, #PIPA511887, L/N SH2420723H and Santa Cruz Biotechnology, sc-23927, I0418, L1818, J1519), anti-vimentin (1:50, BD Biosciences, #550515, L/N 3214517), anti-pY397-FAK (1:25, Cell Signaling, #3283/6), anti-paxillin (1:50, BD, #610051/720868). For visualization of F-actin, cells were stained with rhodamine-phalloidin. The cellular localization of fluorescently labeled proteins was viewed under a high resolution digital fluorescent microscopy (Leica DMI600) or a Zeiss LSM 880 NLO confocal microscope with Fast Airyscan module (Carl Zeiss Microscopy, LLC, White Plains, NY, USA). The time of image capturing and intensity gaining were optimally adjusted and kept constant for all experiments to standardize the fluorescence intensity measurements among experiments. The relative intensity is calculated using the following formula: intensity of each cell/average intensity of control cells. Imaris 9.7.3 (Bitplane, Oxford Instruments Pls, Abingdon, UK) software was used to render spots and surface. The spots and surface modules used an algorithm to create 3D-reconstructed objects based upon fluorescent intensity and quality of rendering. Vimentin filaments were 3D-rendered using the FilamentTracer application on Imaris followed by length measurement. BoundingBox OO length-A measures the length of the shortest principal axis. BoundingBox OO length-B measures the length of the second longest principal axis. BoundingBox OO length-C measures the length of the longest principal axis inside of the object. 

### 2.4. Generation of Stable KD Cells

Stable nestin KD cells were generated using lentiviruses encoding target shRNA as previously described. [23,34]. Briefly, lentiviruses encoding nestin shRNA (sc-36032-V) and control shRNA (sc-108080) were purchased from Santa Cruz Biotechnology. Human ASM cells were infected with control shRNA lentiviruses or target shRNA lentiviruses for 12 h followed by 3–4 day culture. We used puromycin to select positive clones expressing shRNAs. The expression levels of target proteins in these cells were assessed by immunoblot analysis and/or immunofluorescence microscopy. KD cells and control cells were stable at least five passages after initial infection. 

### 2.5. Wound Healing Assay

The wound healing assay can be used to determine directed cell migration [1,35]. An artificial wound was made in the monolayer of control and nestin KD cells by scraping a 10 μL pipette tip across the bottom of the dish. Cells in the medium containing 10% FBS were allowed to migrate for 12 h in a 37 °C incubation chamber with 5% CO_2_. Cell images were taken using a microscope. The remaining open area of the wound was measured using the NIH ImageJ software (Bethesda, MD, USA). 

### 2.6. Time-Lapse Microscopy

Cells were plated in 6-well culture plates with Ham’s F12 medium supplemented with 10% FBS and cultured in a CO_2_ incubator for 4 h to reach 20–30% of confluence. Culture plates were then placed in a stage incubator filled with 5% CO_2_ at 37 °C. Cell migration was monitored every 10 min for 12 h using a Leica DMI600 microscope system (Deerfield, IL, USA). A 10×/dry phase-contrast objective was used for image acquisition. We used the NIH ImageJ software to quantitatively assess accumulated distance, Euclidean distance, and speed of cell migration.

### 2.7. Statistical Analysis

All statistical analysis was performed using Prism software (GraphPad Software, San Diego, CA, USA). Differences between pairs of groups were analyzed by Student’s *t*-test. A comparison among multiple groups was performed by one-way or two-way ANOVA followed by a post hoc test (Tukey’s multiple comparisons). Values of n refer to the number of experiments used to obtain each value. *p* < 0.05 was considered to be significant.

## 3. Results

### 3.1. Nestin Promotes Migration of Smooth Muscle Cells

As the role of nestin in migration is cell-type dependent [9,10], we determined whether nestin affects the migration of human ASM cells. We used lentivirus encoding shRNA against nestin to reduce nestin expression in cells. Immunoblot analysis verified lower nestin protein expression in cells expressing nestin shRNA as compared to cells expressing control shRNA (Figure 1A). We utilized time-lapse microscopy [1,35,36] to assess the role of nestin in migration. The accumulated distance, Euclidean distance, and speed of nestin KD cells were reduced as compared to control cells (Figure 1B–E). We also used the wound healing assay to assess the effects of nestin KD on cell motility. The opening area in nestin KD cells after 12 h was greater compared to control cells (Figure 1F). These results suggest that nestin plays a positive role in regulating ASM cell migration.

### 3.2. Nestin KD Affects the Assembly of Focal Adhesions

We also noticed that nestin KD reduced cell spreading as evidenced by reduced cell body area in nestin KD cells (Appendix A). As focal adhesions are important for cell migration, we determined whether nestin KD affects the morphology of focal adhesions. Paxillin is a marker of focal adhesions [1,15,37] whereas pY397-FAK is an indication for the initial activation of FAK, which is phosphorylated upon integrin activation [38]. Thus, we immunostained cells for these two proteins, and visualized cells by confocal microscopy followed by image analysis using the Imaris software. Majority of paxillin staining colocalized with pY397-FAK labeling in cell periphery (Figure 2A, white arrows). However, paxillin was not found in some of peripheral pY397-FAK (Figure 2A, cyan arrows). The results are consistent with the concept that pY397-FAK induces the initial activation of FAK, which recruits paxillin to focal contacts [2,37,38,39].

More importantly, we noticed that nestin KD reduced the area of paxillin labeling in cells (Figure 2A,B). Interestingly, nestin KD did not affect the length of smaller paxillin labeling (Figure 2C,D). However, nestin KD diminished the length of larger paxillin cluster (Figure 2E). Similarly, nestin KD reduced the area of pY397-FAK larger clusters and the length of larger pY397-FAK staining (Figure 2F–I). Moreover, the expression of paxillin and pY397-FAK was not affected by nestin KD (Appendix A). These results suggest that nestin regulates the formation of focal adhesions during cell spreading. 

### 3.3. Nestin KD Inhibits Phosphorylation of Plk1 and Vimentin

As both nestin and Plk1 are able to focal adhesion assembly (Figure 2) [15], we questioned whether nestin plays a role in affecting Plk1 phosphorylation at Thr-210, an indication of Plk1 activation [14,15,40]. We evaluated the effects of nestin KD on Plk1 phosphorylation at this residue. Plk1 phosphorylation at Thr-210 was reduced in nestin KD cells as evidenced by immunoblot analysis (Figure 3A). In addition, vimentin is one of the major targets of Plk1 [14,15,41], we also determined the role of nestin in vimentin phosphorylation at Ser-56 using immunoblot analysis. Nestin KD attenuated vimentin phosphorylation at Ser-56 in human ASM cells (Figure 3B). 

### 3.4. Nestin Regulates Connection of Vimentin Filaments with Paxillin

As vimentin phosphorylation at Ser-56 regulates the spatial distribution of the vimentin network [12,13,15,33], we determined whether nestin KD affects the vimentin framework using confocal microscopy and the Imaris software. We noticed that the length of vimentin filaments was shorter in nestin KD cells (Figure 3C). Quantification analysis showed that the length of vimentin filaments was reduced to 37% in nestin KD cells (Figure 3D). Moreover, 60% of paxillin clusters were close to vimentin filaments in control cells, which was reduced by nestin KD (Figure 3E). This was verified by quantitative analysis of paxillin clusters that were away from the filaments (Figure 3F). 

### 3.5. Vimentin Phosphorylation-Mimic Mutant Enhances Migration, Vimentin Reorganization, and Focal Adhesion Size of Nestin KD Cells

To further assess the role of vimentin phosphorylation at Ser-56 in the process, we evaluated the effects of the vimentin phosphorylation-mimic mutant S56D on migration, the vimentin network, and focal adhesions of nestin KD cells. As a control, we also assessed the effects of the non-phosphorylatable vimentin mutant S56A. We noticed that the migratory capabilities of nestin KD cells expressing S56A were relatively slow (Figure 4A–C). In contrast, the expression of S56D vimentin enhanced accumulated distance, Euclidean distance, and speed of nestin KD cells (Figure 4A–C). The wound healing assay also showed that S56D vimentin increased the migration of nestin KD cells (Appendix A). Moreover, the expression of S56D vimentin enhanced filament length and contacts of the filaments to paxillin as compared to S56A mutant (Figure 4D–G). Furthermore, S56D vimentin increased focal adhesion sizes of nestin KD cells (Figure 5A–J) and cell spreading (Appendix A). These results suggest that vimentin phosphorylation at this position is important for nestin-regulated migration, rearrangement of the vimentin framework, and focal adhesion morphology. 

### 3.6. Nestin KD Does Not Affect Localization of c-Abl and Cortactin at the Cell Edge

As c-Abl and cortactin are recruited to the cell edge to promote cell protrusion [2,35,42], we determined whether nestin has a role in the recruitment of c-Abl and cortactin. Interestingly, nestin KD did not significantly influence the localization of c-Abl and cortactin at the cell edge (Figure 6A). 

### 3.7. Nestin KD Does Not Affect F-Actin and Myosin Light Chain Phosphorylation

As described earlier, myosin activity increases during cell migration, which facilitates the retraction of the rear. Thus, we evaluated whether nestin regulates myosin light chain phosphorylation at Ser-19, an indication of myosin activation. We found that nestin KD did not affect myosin light chain phosphorylation at this residue (Figure 6B). Moreover, F-actin at the cell edge and stress fibers in lamellipodia are critical for directed cell migration [2,4,36]. We also assessed the effects of nestin KD on F-actin in spreading cells. The intensity of F-actin at the cell edge and stress fibers were not affected by nestin KD (Figure 6C–E). 

## 4. Discussion

Nestin is a class VI IF protein that is implicated in the proliferation/regeneration of neurons, cancer cells, and skeletal muscle [6,7,8]. The role of nestin in cell migration is not fully understood. In this study, we discover that nestin positively modulates ASM cell migration. Moreover, we present evidence that nestin regulates cell migration by controlling the Plk1-vimentin-focal adhesion pathway. 

Focal contact assembly is critical for directed cell migration [1,2,43]. In this study, the majority of paxillin clusters were colocalized with pY397-FAK labeling in the cell periphery. However, paxillin was not present in some of the peripheral pY397-FAK. The results are consistent with the concept that pY397-FAK induces the initial activation of FAK, which recruits paxillin to focal contacts [2,37,38,39]. More importantly, nestin deficiency reduced the size of focal adhesions, particularly that of larger focal adhesions in cells. The results indicate that nestin is important for focal adhesion assembly. Interestingly, nestin KD of PC-3 prostate cancer cells by siRNA increases focal adhesion size and invasion by affecting the distribution of phosphorylated FAK and integrin α5 [10]. These studies suggest that cancer cells and noncancer cells may utilize different cellular processes to control migration and invasion.

In this study, nestin KD inhibited Plk1 phosphorylation at Thr-210. The results suggest that nestin plays a positive role in regulating Plk1 activity. Vimentin is one of the major substrates of Plk1; Plk1 catalyzes vimentin phosphorylation at Ser-56 [14,15,41]. Vimentin phosphorylation has been shown to induce partial vimentin disassembly and the spatial reorientation of the vimenitn network in cells in response to chemical stimulation [33]. Although we do not know whether vimentin phosphorylation affects its disassembly during migration, vimentin phosphorylation at this position regulates the spatial organization of the vimentin network in motile cells [15]. Here, we found that nestin KD inhibited vimentin phosphorylation at this residue and vimentin network organization. Moreover, nestin KD reduced the contacts of vimentin filaments with focal adhesions and focal adhesion size. This finding is supported by the notion that the connection of vimentin filaments with focal adhesions affects focal adhesion morphology [11,13]. Therefore, we propose that nestin regulates the phosphorylation of Plk1 and vimentin. The phosphorylation of vimentin at Ser-56 facilitates the reorganization of the vimentin network, focal adhesion assembly, and migration. 

It is currently unknown how vimentin phosphorylation and the attachment of vimentin filaments to focal adhesions regulate focal contact formation. Interestingly, vimentin filaments recruit the Rac-GEF VAV2 to focal adhesions to promote FAK activation and focal adhesion assembly in nonmuscle cells [11]. The PKCε-mediated phosphorylation of vimentin increases integrin translocation to the plasma membrane [11,44]. Therefore, it is likely that phosphorylation at Ser-56 regulates the connection of the vimentin network with focal adhesions, which affects VAV2 activation, integrin recruitment, and focal adhesion formation. 

c-Abl and cortactin participate in the regulation of cell migration [2,39,45,46]. In this study, nestin deficiency did not affect the recruitment of c-Ab and cortactin to the cell edge. Their recruitment may be orchestrated by integrins during migration [35,42]. Furthermore, nestin KD did not influence F-actin and myosin light chain phosphorylation at Ser-19. These results suggest that the nestin network is not involved in the regulation of the dynamic actin cytoskeleton and myosin activation of spreading/motile cells.

As described earlier, ASM cell migration plays a role in progression of airway remodeling, a hallmark of allergic asthma [1,2,3]. In this study, we demonstrate that nestin positively regulates ASM cell migration. This raises the possibility that nestin may participate in the progression of airway remodeling. Future studies are required to test this possibility. 

In conclusion, we uncover a novel role for nestin in cell migration. During migration, nestin regulates the phosphorylation and activation of Plk1, which mediates the phosphorylation of vimentin at Ser-56. The phosphorylation of vimentin at this residue promotes the connection of vimentin filaments with paxillin and focal adhesion assembly, which enhances cell migration (Figure 7). 

## Figures and Tables

**Figure 1 cells-11-03047-f001:**
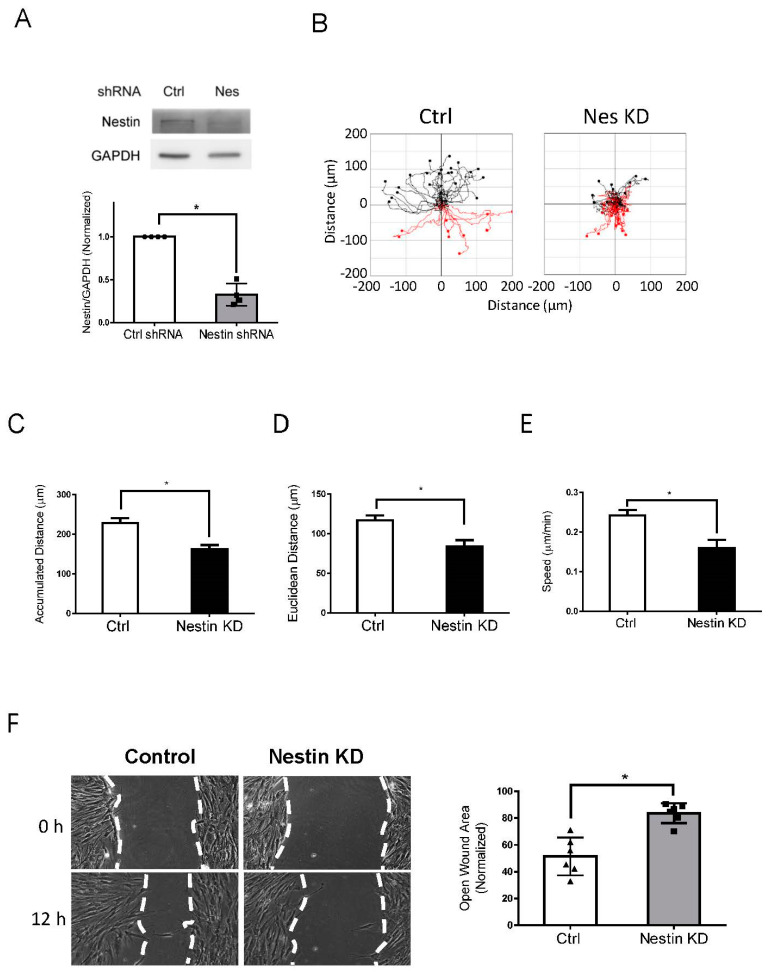
Knockdown (KD) of nestin reduces human ASM cell migration. (**A**) Protein expression of human ASM cells stably expressing control (Ctrl) shRNA or nestin shRNA was evaluated by immunoblotting. Data are mean values of experiments from 4 batches of cell culture. Error bars indicate SE. (**B**) Cell migration was tracked using a time-lapse microscope. Images were taken every 10 min for 12 h. Migration plots (32 cells from each group) were generated using the NIH ImageJ software. Red tracks indicate downward migration whereas black tracks indicate upward migration. (**C**–**E**) Nestin (Nes) KD reduced accumulated distance, Euclidean distance, and speed of cell migration. (**F**) Nestin KD inhibits the migration of human ASM cells as evidenced by the wound healing assay, which was described in detail in Material and Methods. Data are mean values of six experiments. Error bars indicate SE. * *p* < 0.05. *t*-test was used for statistical analysis.

**Figure 2 cells-11-03047-f002:**
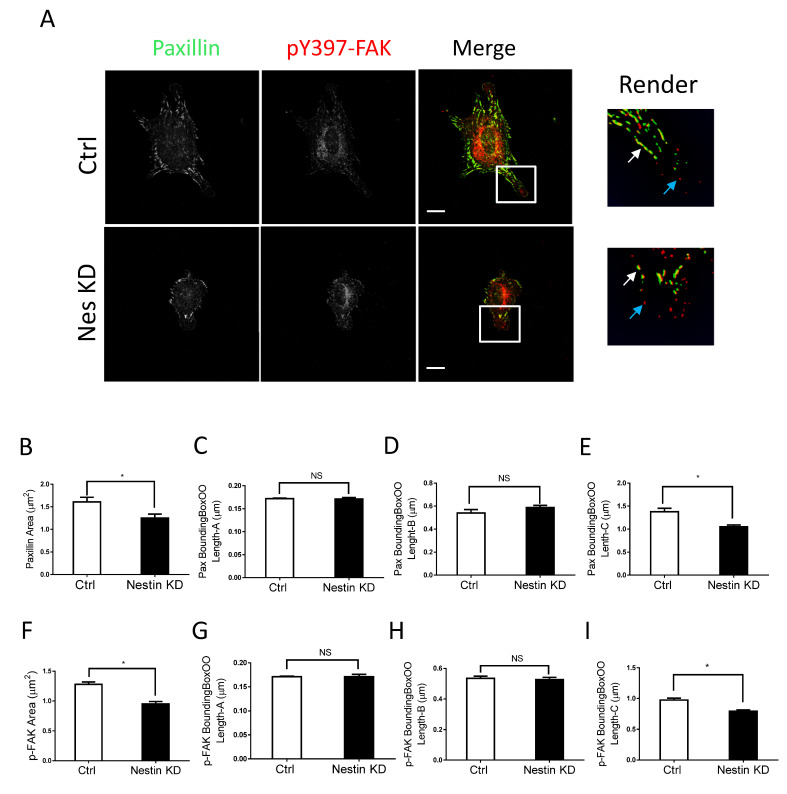
Nestin KD reduces size of focal adhesions. (**A**) Ctrl and nestin KD human ASM cells were plated on collagen-coated coverslips for 45 min and immunostained for paxillin and pY397-FAK. Cell images were captured using a confocal microscope and the Imaris software was used to render and analyze images. Scale bar, 15 µm. The white arrows point spots colocalized with paxillin and pY397-FAK. The cyan arrows point to pY397-FAK labeling spots. (**B**–**I**) The area, BoundingBoxOO Length-A/B/C of paxillin and pY397-FAK staining were calculated using the Imaris software. BoundingBox OO length-A measures the length of the shortest principal axis. BoundingBox OO length-B measures the length of the second longest principal axis. BoundingBox OO length-C measures the length of the longest principal axis inside of the object. Data are means ± SE (*n* = 5, * *p* < 0.05). *t*-test was used for statistical analysis. NS, not significant.

**Figure 3 cells-11-03047-f003:**
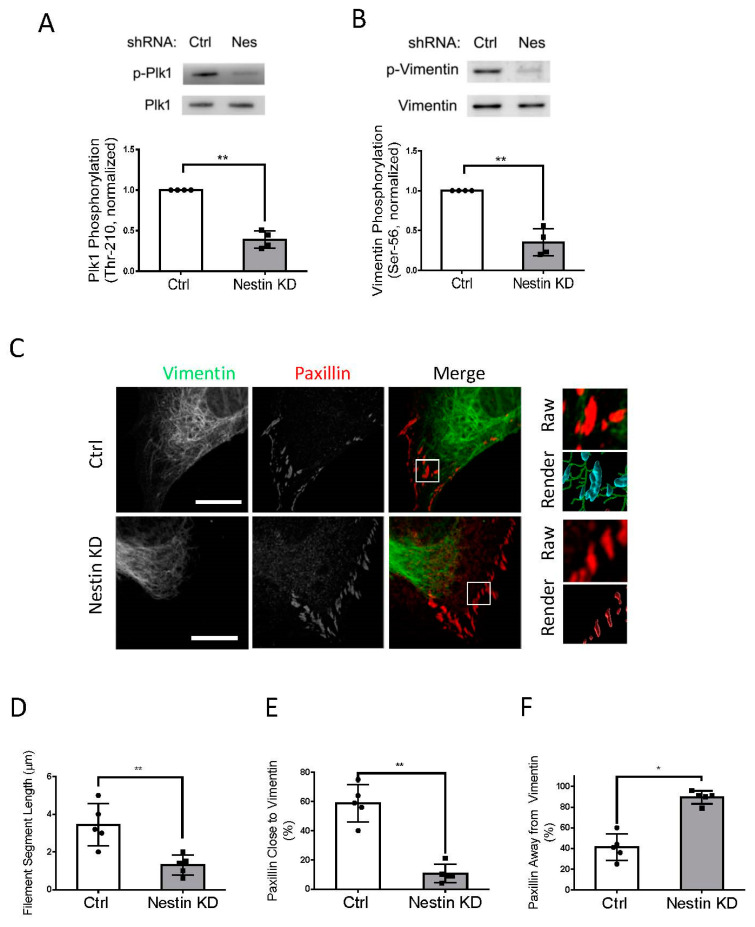
Nestin KD affects Plk1 phosphorylation at Thr-210, vimentin phosphorylation at Ser-56, and vimentin network. (**A**) Extracts of Ctrl and nestin KD cells were immunoblotted with antibodies against phospho-Plk1 (Thr-210) and total Plk1. Plk1 phosphorylation is normalized to the level obtained from Ctrl cells. Data are mean ± SE (*n* = 4). (**B**) Extracts of Ctrl and nestin KD cells were immunoblotted with antibodies against phospho-vimentin (Ser-56) and total vimentin. Vimentin phosphorylation is normalized to the level obtained from Ctrl cells. Data are mean ± SE (*n* = 4). (**C**) Ctrl and nestin KD ASM cells were plated on collagen-coated coverslips for 45 min and immunostained for vimentin and paxillin. Cell images were taken using a Zeiss LSM880 microscope with Airyscan. The images of vimentin filaments and paxillin staining in cell protrusions were used for Imaris quantitative analysis. Scale bar: 15 μm. Imaris software was utilized to 3D-render vimentin filaments and paxillin surfaces. 3D-rendered vimentin is green, paxillin closed to vimentin is cyan, and paxillin alone is red. (**D**) The Imaris software was utilized to quantify the length of filament segments. (**E**,**F**) The percent of paxillin clusters closed to or away from vimentin filaments was assessed using the Imaris software. Data are mean ± SE (*n* = 5). *t*-test was used for statistical analysis. * *p* < 0.05; ** *p* < 0.01.

**Figure 4 cells-11-03047-f004:**
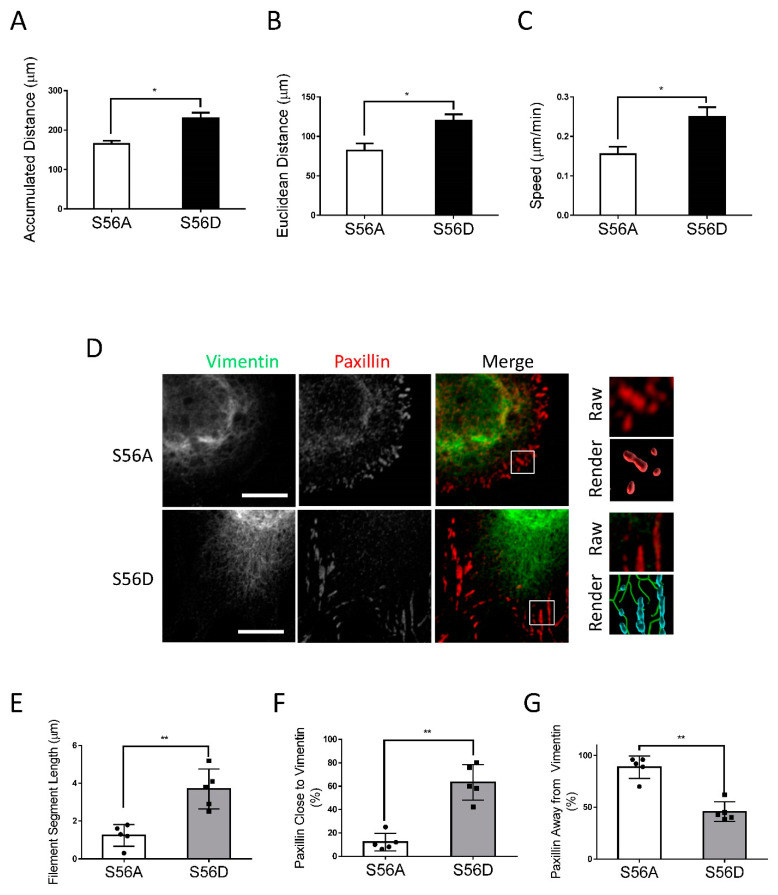
Vimentin S56D affects migration and vimentin network of nestin KD cells. To assess the role of vimentin phosphorylation at Ser-56, nestin KD cells were transfected with S56A or S56D vimentin. Cell migration was evaluated using a time-lapse microscope. (**A**–**C**) S56D, but not S56A, increases accumulated distance, Euclidean distance, and speed of nestin KD cell migration (*n* = 32–34 cells/each group). (**D**–**G**) Cell images were captured and analyzed using the methods described in Figure 3 legend. Expression of S56D in nestin KD cells increases vimentin filament length and the connection of the filaments to paxillin. Data are mean ± SE (*n* = 5). * *p* < 0.05; ** *p* < 0.01. *t*-test was used for statistical analysis. Scale bar, 15 µm.

**Figure 5 cells-11-03047-f005:**
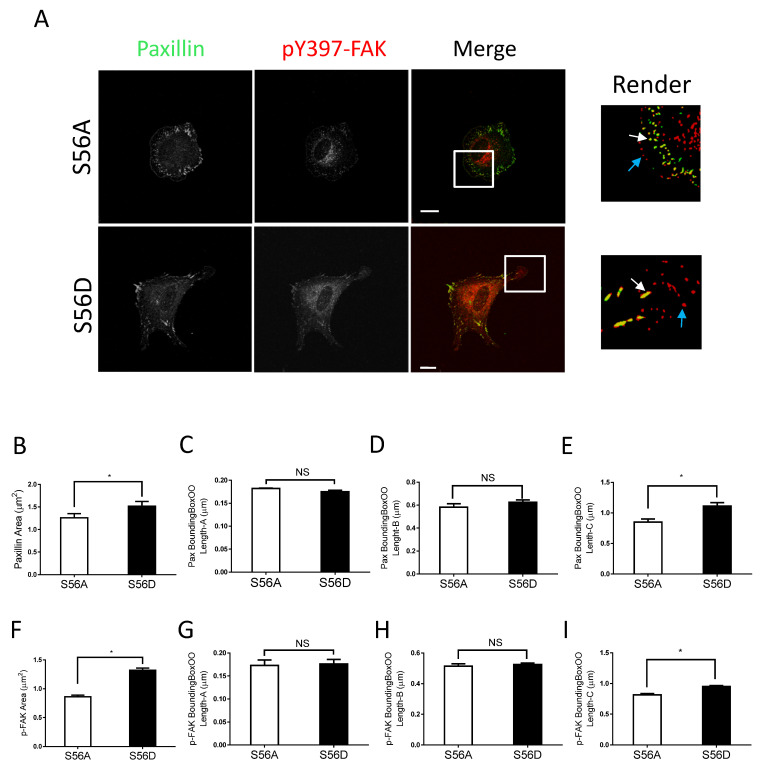
Vimentin S56D regulates focal adhesions. (**A**) Nestin KD human ASM cells were transfected with S56A or S56D vimentin, and cell images were collected and analyzed using the methods described in Figure 2 legend. Scale bar: 15 μm. The white arrows point spots colocalized with paxillin and pY397-FAK. The cyan arrows point to pY397-FAK labeling spots. (**B**–**I**) The area, BoundingBoxOO Length-A/B/C of paxillin and pY397-FAK staining were calculated using the Imaris software. S56D increases the areas and BoundingBox OO length-C of paxillin and pY397-FAK. Data are means ± SE (*n* = 5, * *p* < 0.05). *t*-test was used for statistical analysis. NS, not significant.

**Figure 6 cells-11-03047-f006:**
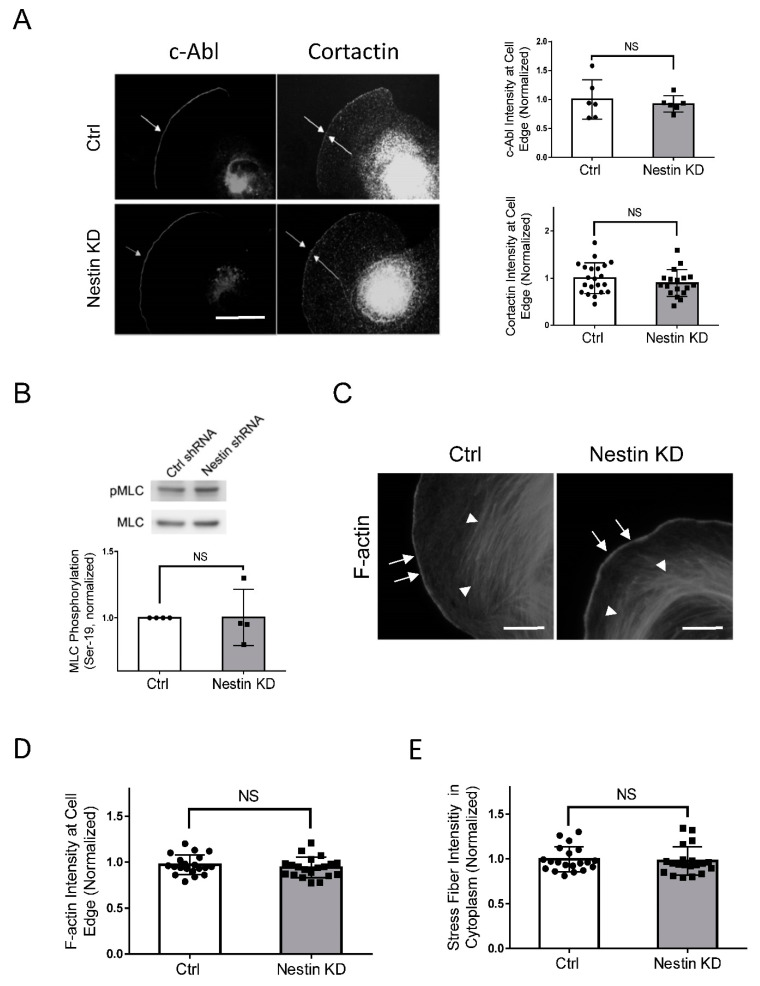
Nestin KD does not affect the recruitment of c-Abl and cortactin, myosin phosphorylation, and F-actin. (**A**) Ctrl and nestin KD human ASM cells were plated on collagen-coated coverslips for 45 min and immunostained for c-Abl and cortactin, Scale bar: 10 µm. The arrows point to the cell edge. The relative intensity of c-Abl and cortactin was calculated using the following formula: intensity of each cell/average intensity of control cells. Data are mean values of at least 20 cells from three experiments. Error bars indicate SE. NS, not significant. (**B**) Myosin light chain phosphorylation at Ser-19 of Ctrl and nestin KD human ASM cells was evaluated by immunoblot analysis. The phosphorylation levels of nestin KD cells are normalized to Ctrl cells. Data are mean values of 4 batches of cell cultures. Error bars indicate SE. (**C**–**E**) Ctrl and nestin KD human ASM cells were plated on collagen-coated coverslips for 45 min and stained with phalloidin for F-actin. Scale bar: 10 µm. The arrows point to the cell edge. The arrow heads point to stress fibers. Data are mean values of 4 batches of cell cultures. Error bars indicate SE. NS, not significant.

**Figure 7 cells-11-03047-f007:**
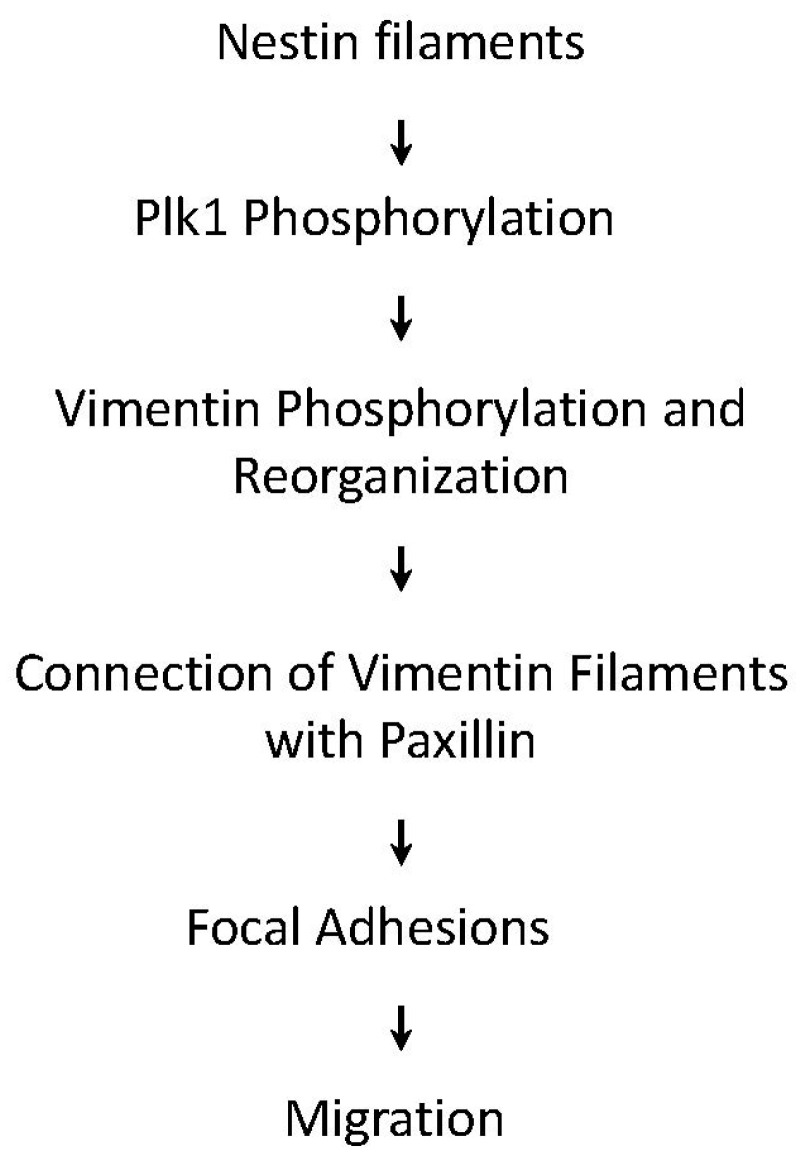
Proposed mechanism: During migration, nestin regulates the activation of Plk1, which mediates the phosphorylation of vimentin at Ser-56. The phosphorylation of vimentin promotes the connection of vimentin filaments with paxillin, focal adhesion assembly, and cell migration.

## Data Availability

Essential datasets supporting the conclusions are included in this published article.

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
