# Peer review of "Nestin Modulates Airway Smooth Muscle Cell Migration by Affecting Spatial Rearrangement of Vimentin Network and Focal Adhesion Assembly"

_cells, 2022, doi:10.3390/cells11193047_

Round 1
Reviewer 1 Report
I have carefully reviewed the manuscript entitled “Nestin modulates airway smooth muscle cell migration by affecting spatial rearrangement of vimentin network and focal adhesion assembly Special Issue: Airway Smooth Muscle and Asthma” by Wang and colleagues.
The study is very interesting and well conducted, the sophisticated design allows to deeply explore the scientific question at the base of its aim, the results and conclusions are consequently soundly based and reliable.
In my opinion, there are just some points of the manuscript that need to be reorganized to improve its readability and to better put in light the findings.
I think that the study design should be better and deeper presented at the end of the introduction. In its current form, the design is only briefly mentioned, and reading the M&M and, in particular, the results section, the reader have to face new information about analyses that are not mentioned before in the text. Moreover, the rationale at the base of some choices should be better explained (e.g. why did you decide to study the Plk1 in relation to nestin. This point is only briefly mentioned at the beginning of the paragraph 3.3, it deserves some more mines in the introduction).
In the discussion, the relationship between the novel findings on nestin role and the ASM cell migration should be better discussed. Globally, I found the discussion quite limited to the single findings, while the aim stated at beginning of the manuscript concerned the ASM cell migration. Thus, a larger contextualization of the findings on this sense could better put in light the study.
Other minor point that I suggest to address are reported here below:
- I’m not familiar with the journal guidelines, but I found the abstract quite uncommonly redacted.
- M&M, paragraph 2.5: how have you selected this model of wound healing essay? I possible, had the citation or at least an explanation.
- Same paragraph: please add some explanation about the method used to measure the area (which ImageJ plugin? Which procedure?). I will help the reader to understand the reliability of the values and also to reproduce this analysis.
- M&M, paragraph 2.7: please explain if and how do you calculate the distribution of your data.
Author Response
We thank the reviewer for the constructive criticisms and suggestions. We have revised the manuscript to address these concerns and suggestions. Below are detailed our responses:
Comment 1: “I think that the study design should be better and deeper presented at the end of the introduction. In its current form, the design is only briefly mentioned, and reading the M&M and, in particular, the results section, the reader have to face new information about analyses that are not mentioned before in the text. Moreover, the rationale at the base of some choices should be better explained (e.g. why did you decide to study the Plk1 in relation to nestin. This point is only briefly mentioned at the beginning of the paragraph 3.3, it deserves some more mines in the introduction).”
Response 1: Thanks for the comments. We have added more information about the experimental design in the Results and Figure Legends, which is a standard format for a cellular article. With regard to why we choose Plk1, this is because nestin KD reduces focal adhesion assembly. Plk1 has been previously shown to regulate focal adhesion assembly.
Comment 2: “In the discussion, the relationship between the novel findings on nestin role and the ASM cell migration should be better discussed. Globally, I found the discussion quite limited to the single findings, while the aim stated at beginning of the manuscript concerned the ASM cell migration. Thus, a larger contextualization of the findings on this sense could better put in light the study.”
Response 2: The reviewer is right. We have revised the section of Discussion. Particularly, we rephrase the last sentence of Discussion.
Comment (Minor) 3: “I’m not familiar with the journal guidelines, but I found the abstract quite uncommonly redacted.”
Response 3: The abstract includes rationale, major methods, results, and conclusion, which is consistent with the guideline.
Comment (Minor) 4: “M&M, paragraph 2.5: how have you selected this model of wound healing essay? I possible, had the citation or at least an explanation.”
Response 4: The wound healing assay is a standard method to determine cell migration in vitro. Particularly, it can be used to determine directed cell migration.
Comment (Minor) 5: “Same paragraph: please add some explanation about the method used to measure the area (which ImageJ plugin? Which procedure?). I will help the reader to understand the reliability of the values and also to reproduce this analysis.”
Response 5: We tried some plugins like “wound healing size tool”. But, it did not work well. We just find edges manually because we do not have many samples.
Comment (Minor) 6: “M&M, paragraph 2.7: please explain if and how do you calculate the distribution of your data.”
Response 6: Thanks for the question. We did not calculate the distribution of our data. Empirically, these cellular and biochemical data are normal distribution. We used Graphpad to perform statistical analysis.
Reviewer 2 Report
Wang et al., investigated the role of nestin in migration of airway smooth muscle cell migration which is important in airway re modelling in allergic asthma and COPD. The manuscript is well written and the authors presented their experiments and results in a simple way. The only thing is that the authors need to expand the discussion.
Author Response
We thank the reviewer for the constructive suggestion. We have revised the manuscript to address the suggestion. Below are detailed our responses:
Comment 1: “Wang et al., investigated the role of nestin in migration of airway smooth muscle cell migration which is important in airway remodelling in allergic asthma and COPD. The manuscript is well written and the authors presented their experiments and results in a simple way. The only thing is that the authors need to expand the discussion..”
Response 1: Thanks for the positive comment. We have expanded the discussion. Particularly, rephrase the last sentence and add a paragraph to discuss potential role of nestin in airway remodeling.